# Localization with Sampling-Argmax

**Jiefeng Li   Tong Chen   Ruiqi Shi   Yujing Lou   Yong-Lu Li   Cewu Lu**
Shanghai Jiao Tong University
{ljf_likit,chentong1023, gzfoxie, louyujing, yonglu_li, lucewu}@sjtu.edu.cn

## Abstract

Soft-argmax operation is commonly adopted in detection-based methods to localize the target position in a differentiable manner. However, training the neural network with soft-argmax makes the shape of the probability map unconstrained. Consequently, the model lacks pixel-wise supervision through the map during training, leading to performance degradation. In this work, we propose sampling-argmax, a differentiable training method that imposes implicit constraints to the shape of the probability map by minimizing the expectation of the localization error. To approximate the expectation, we introduce a continuous formulation of the output distribution and develop a differentiable sampling process. The expectation can be approximated by calculating the average error of all samples drawn from the output distribution. We show that sampling-argmax can seamlessly replace the conventional soft-argmax operation on various localization tasks. Comprehensive experiments demonstrate the effectiveness and flexibility of the proposed method. Code is available at https://github.com/Jeff-sjtu/sampling-argmax.

## 1   Introduction

Localizing the target position from the input is a fundamental task in the field of computer vision. Common approaches to localization can be divided into two categories: regression-based and detection-based. Detection-based methods show superiority over regression-based methods and demonstrate impressive performance on a wide variety of tasks [41, 34, 40, 7, 15, 9, 17, 12, 32, 18, 31]. Probability maps (also referred to as heat maps) are predicted in detection-based methods to indicate the likelihood of the target position. The position with the highest probability is retrieved from the probability map with the *argmax* operation. However, the argmax operation is not differentiable and suffers from quantization error. For accurate localization and end-to-end learning, *soft-argmax* [4, 3] is proposed as an approximation of argmax. It has found a wide range of applications in human pose estimation [34, 21, 22, 35], facial landmark localization [9, 20, 1], stereo matching [41, 13, 2] and object keypoint estimation [31].

Nevertheless, the mechanism of training networks with soft-argmax is rarely studied. The conventional training strategy is to minimize the error between the output coordinate from soft-argmax and the ground truth position. However, this strategy is deficient since it only provides constraints to the expectation of the probability map, not to its shape. As shown in Figure 1, these two maps have the same mean values, but the bottom one is more concentrated. In well-calibrated probability maps, positions that locate closer to the ground truth have higher probabilities. Reliable confidence scores of localization results could be provided, which is essential in unconstrained real-world applications and downstream tasks. Besides, imposing constraints on the probability map can provide supervised pixel-wise gradients and facilitate the learning process.

Prior work [28] attempts to shape the probability map by introducing hand-crafted regularizations. The variance regularization encourages the variance of the probability map to get close to the pre-defined variance. The Gaussian regularization forces the probability map to resemble a Gaussian distribution. We argue that these variants are overconstrained. The hand-crafted constraints are not

35th Conference on Neural Information Processing Systems (NeurIPS 2021).

always correct in different cases. For example, the underlying shape of the probability map is not necessarily Gaussian, and the underlying variance might change as the input changes. Imposing the model to learn a fixed-variance Gaussian distribution might degrade the model performance.

In this work, we present *sampling-argmax*, a novel training method to obtain well-calibrated probability maps and improve the localization accuracy. To constrain the shape of the map, we replace the objective function of minimizing "the error of the expectation" with minimizing "the expectation of the error". In this way, the network is encouraged to generate higher probabilities around the ground truth position.

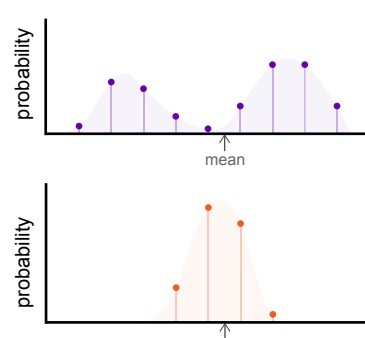

A natural way to estimate the expectation is by calculating the probability-weighted sum of the errors at all grid positions. However, we find that the gradient has high variance, and the model is hard to train. To address this issue, we choose to approximate the expectation by sampling. The expectation of the error is calculated as the mean error of all samples. Therefore, the sampling process should be differentiable for end-to-end learning.

Figure 1: **Top**: an unconstrained probability map. **Bottom**: a well-calibrated probability map. These two maps have different shapes but a same mean value.

In our work, we show that the likelihood of the target position can be modelled in the continuous space with a mixture distribution. Samples can be drawn from the mixture distribution by three steps: i) generate categorical weights of the mixture distribution from the probability map; ii) draw samples from sub-distributions; iii) obtain a sample by the category-weighted sum. The benefit of using mixture distribution is that differentiable sampling from arbitrary continuous distributions can be resolved by differentiable sampling from categorical distributions, which is less challenging and can be addressed by off-the-shelf discrete sampling methods.

Sampling-argmax is simple and effective. With out-of-the-box settings, it can be integrated into methods that using soft-argmax operation. To study its effectiveness, we conduct experiments on a variety of localization tasks. Quantitative results demonstrate the superiority of sampling-argmax against soft-argmax and its variants. In summary, the contributions of this work are threefold:

- We propose *sampling-argmax* for improving detection-based localization methods. By minimizing "the expectation of the error", the network generates well-calibrated probability maps and obtains higher localization accuracy.
- We show the output likelihood can be formulated as a mixture distribution and develop a differentiable sampling pipeline.
- Comprehensive experiments show that sampling-argmax is effective and can be flexibly generalized to different localization tasks.

## 2 Preliminary

Given a learned discrete probability map $\pi$, the value $\pi_{y_i}$ indicates the probability of the predicted target appearing at $y_i$. A direct way to localize the target is taking the position with the maximum likelihood. However, this approach is non-differentiable, and the output is discrete, which impedes end-to-end training and brings quantization errors. Soft-argmax is an elegant approximation to address these issues:

$$\hat{y} = \texttt{soft-argmax}(\pi) = \sum_i \pi_{y_i} y_i. \tag{1}$$

Notice that $\pi$ is a normalized distribution and the soft-argmax operation calculates the probability-weighted sum, which is equivalent to taking the expectation of the probability map $\pi$. A conventional way to train the model with the soft-argmax operation is minimizing the distance between the expectation and the ground truth:

$$\mathcal{L} = d(y_t, \mathbb{E}_y[y]) \approx d(y_t, \sum_i \pi_{y_i} y_i), \tag{2}$$

where $y_t$ denotes the ground truth position and $d(\cdot, \cdot)$ denotes the distance function, e.g. $\ell_1$ distance. We refer to this objective function as "the error of the expectation".

# 3 Method

The conventional detection-based method with soft-argmax only supervises the expectation of the probability map. The shape of the distribution remains unconstrained. In well-calibrated probability maps, the positions closer to the ground truth should have higher probabilities. To this end, we proposed a new objective function that optimizes "the expectation of the error" instead of "the error of the expectation". In particular, the objective function is formulated as:

$$\mathcal{L} = \mathbb{E}_y[d(y_t, y)]. \tag{3}$$

The learned distribution tends to allocate high probabilities around the ground truth to minimize the entire loss. In this way, the shape of the probability map is implicitly constrained.

**Discrete Distribution.** The probability map $\pi$ predicted by the neural network is discrete. Similar to the soft-argmax operation, the expectation of error can be approximated by calculating the probability-weighted sum of the errors at all grid positions:

$$\mathcal{L} = \mathbb{E}_y[d(y_t, y)] \approx \sum_i \pi_{y_i} d(y_t, y_i). \tag{4}$$

This approximation treats the distribution of the target position as a discrete distribution. The target only appears at the grid positions, i.e. at position $y_i$ with the probability $\pi_{y_i}$.

However, because the underlying target lies in a continuous space, modelling the distribution as a discrete distribution is not accurate. The probability map has limited resolution due to the computation complexity. Besides, we find the model is slow to converge by training with Equation 4. When training with Equation 4, the model only obtains 30.9 mAP on COCO Keypoint, while conventional soft-argmax obtains 64.5 mAP. For analysis, we derive the gradient from the loss function to the model parameters $\theta$ under the discrete approximation:

$$\begin{aligned}
\nabla_\theta \mathcal{L} &= \nabla_\theta \mathbb{E}_y[d(y_t, y)] \\
&= \sum_i d(y_t, y_i)\nabla_\theta \pi_{y_i} = \sum_i d(y_t, y_i)\pi_{y_i}\nabla_\theta \log \pi_{y_i} \\
&= \mathbb{E}_y[d(y_t, y)\nabla_\theta \log \pi_y].
\end{aligned} \tag{5}$$

Notice that the form of the gradient is similar to the score function estimator (SF), which is alternatively called the REINFORCE estimator [36]. SF estimator is known to have very high variance and is slow to converge. Therefore, using the discrete approximation for training is not a good solution. This challenge prompts us to explore a better approximation to calculate the expectation of the error.

In the following parts, we present sampling-argmax to estimate the expectation of the error by sampling. We first develop a continuous approximation to the distribution of the target position (Section 3.1). Then we propose a differentiable sampling method (Section 3.2).

## 3.1 Continuous Mixture Distribution

A differentiable process is necessary to estimate the expectation by sampling. However, since the underlying probability density functions can vary among different input images, it is challenging to draw samples from arbitrary distributions differentiably. In this work, we present a unified method by formulating the target distribution as a mixture distribution.

Let $p(y)$ denotes the underlying density function of the target position, which is defined within the boundary of the input image, i.e. $y \in [0, W]$. As illustrated in Figure 2(a), the interval $[0, W]$ can be divided into $n$ subintervals. The density function can be partitioned into shapes in the subintervals. We could use regular shape (rectangles, triangles, Gaussian functions) in subintervals to form the entire function (as illustrated in Figure 2(b-c)).

Formally, given a finite set of probability density functions $\{f_1(y), f_2(y), \cdots, f_n(y)\}$ and weights $\{w_1, w_2, \cdots, w_n\}$ such that $w_i \geq 0$ and $\sum w_i = 1$, the mixture density function $p(y)$ is formulated as a sum:

$$p(y) = \sum_{i=1}^{n} w_i f_i(y). \tag{6}$$

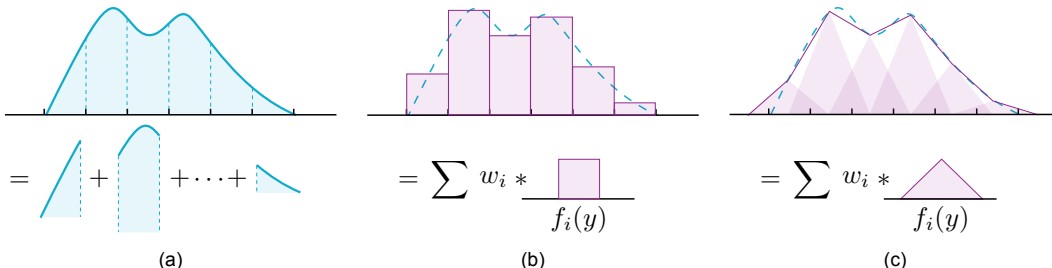

Figure 2: **Representing the continuous distribution as a mixture distribution.** (a) The original probability density function can be viewed as the sum of $n$ sub-functions. Each sub-function can be replaced by standard density functions with proper weights to approximate the original function. (b) Approximate the original function by replacing the sub-functions with uniform distribution. (c) Approximate the original function by replacing the sub-function with the triangular distribution, which is equivalent to the linear interpolation of the discrete weights.

Here, we can leverage the discrete probability map $\pi$ to represent the mixture weights, i.e. $w_i = \pi_{y_i}$. In the context of signal processing, the original function can be perfectly reconstructed if the sample rate (the distance between two adjacent grid points) satisfies the Nyquist-Shannon sampling theorem. However, in our case, the sub-function $f_i(y)$ must be a probability density function, i.e. it has the non-negative values, and its integral over the entire space is equal to $1$. Therefore, with these restrictions, the original function $p(y)$ cannot be perfectly reconstructed. For approximation, we study three different types of standard density functions below.

**Uniform Basis.** For the uniform basis, the sub-function $f_i(y)$ is a uniform distribution centred at the position $y_i$:

$$f_i(y) = \begin{cases} \frac{1}{c}, & y \in [y_i - \frac{c}{2}, y_i + \frac{c}{2}], \\ 0, & otherwise, \end{cases} \tag{7}$$

where $c$ is the distance between two adjacent grid points.

**Triangular Basis.** For the triangular basis, the sub-function $f_i(y)$ is a triangular distribution:

$$f_i(y) = \begin{cases} \frac{1}{c^2}(y - y_i) + \frac{1}{c}, & y \in [y_i - c, y_i), \\ -\frac{1}{c^2}(y - y_i) + \frac{1}{c}, & y \in [y_i, y_i + c), \\ 0, & otherwise. \end{cases} \tag{8}$$

For all $y$, there exist grid points $y_i$ and $y_{i+1}$ that satisfy $y \in [y_i, y_{i+1}]$. Therefore, we have $p(y) = w_i f_i(y) + w_{i+1} f_{i+1}(y) = \frac{w_{i+1} - w_i}{c^2}(y - y_i) + \frac{w_i}{c}$, which is the linear interpolation of $w_i$ and $w_{i+1}$. In other words, using triangular bases is equivalent to the linear interpolation of the discrete probability map.

**Gaussian Basis.** For the Gaussian basis, $f_i(y)$ is the Gaussian function:

$$f_i(y) = \frac{1}{\sigma\sqrt{2\pi}} \exp\left(-\frac{1}{2}(\frac{y - y_i}{\sigma})^2\right). \tag{9}$$

where $\sigma$ denotes the standard deviation. We set $\sigma = c$ by default in the experiments.

## 3.2   Differentiable Sampling

In this part, we present how to draw a sample from the mixture distribution. We first study the non-differentiable process and then present the differentiable approximation.

**Non-differentiable Process.** As illustrated in Figure 3(a), the non-differentiable sampling process can be divided into two steps: i) determine which sub-distribution the sample comes from; ii) draw a sample from the selected sub-distribution. In the first step, the sub-distribution can be selected by drawing a random variable from a categorical distribution. The categorical distribution is indicated

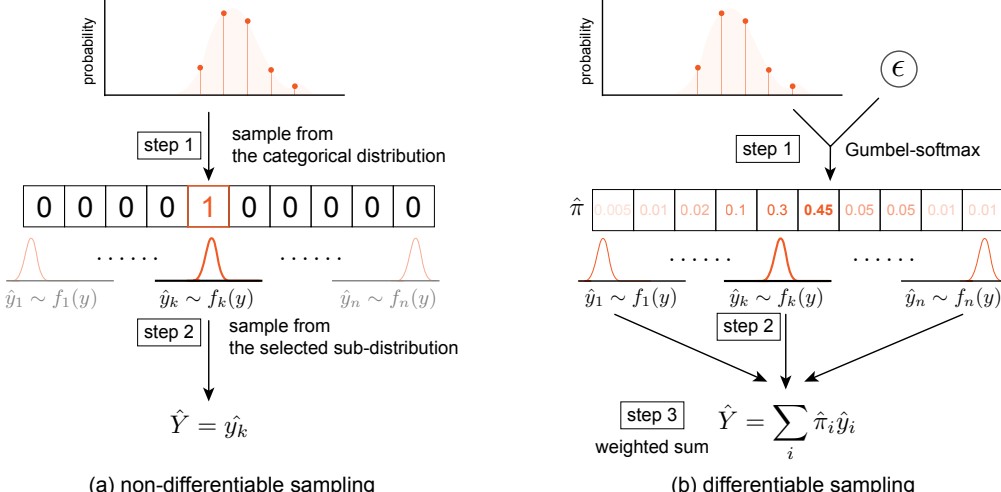

(a) non-differentiable sampling          (b) differentiable sampling

Figure 3: **Illustration of the sampling process.** (a) The non-differentiable process: i) select a sub-distribution by categorical sampling; ii) draw samples from the selected sub-distribution. (b) The differentiable process: i) approximate the categorical sampled weights by Gumbel-softmax; ii) draw samples from all sub-distribution; iii) add all samples together with the sampled weights. Reparameterization allows gradients to flow from the sample to the probability map.

by the predicted probability map $\pi$. The sub-distribution $f_i(y)$ is chosen with the probability $\pi_{y_i}$. There are a number of methods to draw samples from the categorical distribution. Here, we introduce the Gumbel-Max trick [6, 24]:

$$z = \texttt{one\_hot\_max}_i[g_i + \log \pi_i], \qquad (10)$$

where $g_1, \cdots, g_n$ are i.i.d samples drawn from Gumbel(0, 1), and the sample $z$ is a one-hot vector with the value 1 in the maximum categorical column.

In the second step, sampling from the standard basis function is easy to implement. This step is independent of the predicted probability map $\pi$. Therefore, the key to differentiable sampling from the mixture distribution is to make the first step differentiable.

**Differentiable Process.** The differentiable sampling process consists of three steps. In the first step, we adopt the Gumbel-softmax [11] operation to sample the categorical weight from the probability map. Gumbel-softmax is a continuous and differentiable approximation of the Gumbel-Max trick. We can obtain an $(n-1)$-dimensional simplex $\hat{\pi} \in \Delta$:

$$\hat{\pi}_i = \frac{\exp\left((g_i + \log \pi_i)/\tau\right)}{\sum_{k=1}^{n} \exp\left((g_k + \log \pi_k)/\tau\right)}, \qquad (11)$$

where $\hat{\pi} = \{\hat{\pi}_1, \cdots, \hat{\pi}_n\}$ and $\hat{\pi}_i$ denotes the sampled weight of the sub-distribution $f_i(y)$. As the softmax temperature $\tau$ approaches 0, the simplex $\hat{\pi}$ becomes one-hot, and its distribution becomes identical to the categorical distribution $\pi$.

In the second step, we draw a sample $\hat{y}_i$ from every sub-distribution $f_i(y)$. Note that the sampled weight is not completely one-hot. Therefore, we obtain the final sample $\hat{Y}$ in the third step by adding all samples together with the sampled weight $\hat{\pi}$:

$$\hat{Y} = \sum_{i}^{n} \hat{\pi}_i \hat{y}_i. \qquad (12)$$

This process is illustrated in Figure 3(b). With the reparameterization trick, the sample $\hat{Y}$ is computed as a deterministic function of the probability map $\pi$ and the independent random variables. The randomness of the sampling process is transferred to the variable $g_1, \cdots, g_n$. We denote the sampling process as $\hat{Y} = s(\pi, \epsilon)$, where $\epsilon = \{g_1, \cdots, g_n\}$ follows the multivariate Gumbel(0, 1) distribution.

The gradient from the expected error to the model parameters $\theta$ is derived as:

$$\nabla_\theta \mathbb{E}_y[d(y_t, y)] = \nabla_\theta \mathbb{E}_\epsilon[d(y_t, s(\pi, \epsilon))] = \mathbb{E}_\epsilon \left[ \frac{\partial d}{\partial s} \frac{\partial s}{\partial \pi} \frac{\partial \pi}{\partial \theta} \right]. \tag{13}$$

As we see, the gradient of the continuous sampling process is easy to compute via backpropagation. Therefore, we can relax the objective function by calculating the average error of the samples drawn from the mixture distribution. The objective function is written as:

$$\mathcal{L} = \mathbb{E}_{y \sim p(y)}[d(y_t, y)] \approx \frac{1}{N_s} \sum_{k=1}^{N_s} d(y_t, \hat{Y}_k) = \frac{1}{N_s} \sum_{k=1}^{N_s} d(y_t, s(\pi, \epsilon_k)), \tag{14}$$

where $N_s$ denotes the number of samples. In the testing phase, no randomness is introduced, and sampling-argmax degrades to soft-argmax.

While the sampling process is differentiable, the sample $\hat{Y}$ does not follow the original mixture distribution $p(y)$ for non-zero temperature. For small temperatures, the distribution of $\hat{Y}$ is close to $p(y)$, but the variance of the gradients is large. There is a tradeoff between small temperatures and large temperatures. In our experiments, we start at a high temperature and anneal to a small temperature.

## 4   Related Work

**Variants of Soft-Argmax.**   Nibali et al. [28] introduced hand-crafted regularization to constrain the shape of the probability map.

*Variance Regularization.* Variance regularization is to control the variance of the probability map. It pushes the variance of the probability map close to the target variance $\sigma_t^2$:

$$\mathcal{L}_{var} = \|\text{Var}(\pi) - \sigma_t^2\|_2^2, \tag{15}$$

where the target variance $\sigma_t^2$ is a hyperparameter and the variance of the probability map $\text{Var}(\pi)$ is approximated in a discrete manner, i.e. $\text{Var}(\pi) = \sum_i \pi_{y_i}(y_i - \sum_k \pi_{y_k} y_k)^2$.

*Distribution Regularization.* Distribution regularization is to impose strict regularization on the appearance of the heatmap to directly encourage a certain shape. Specifically, [28] forces the probability map to resemble a Gaussian distribution by minimizing the Jensen-Shannon divergence between $\pi$ and target discrete Gaussian distribution:

$$\mathcal{L}_{JS} = D_{JS}(\pi \| \mathcal{N}(\mathbb{E}(y), \sigma_t^2)). \tag{16}$$

Unlike them, our objective function does not set pre-defined hyperparameters for the shape of the map, which makes it general and flexible in applying to various applications.

Other works [12, 15] study how to localize target with soft-argmax in different situations. Joung et al. [12] proposed sinusoidal soft-argmax for cylindrical probabilities map. Lee et al. [15] proposed kernel soft-argmax to make the results less susceptible to multi-modal probability map. Our work is compatible with these methods by applying the sinusoidal function to the grid positions or multiplying the Gaussian kernel before obtaining the probability map.

**Differentiable Sampling.**   Differentiable sampling for a discrete random variable has been studied for a long time. Maddison et al. [23] and Jang et al. [11] concurrently proposed the idea of using a softmax of Gumbel as relaxation for differentiable sampling from discrete distributions. Kočiskỳ et al. [14] relaxed the discrete sampling by drawing symbols from a logistic-normal distribution rather than drawing from softmax. In this work, unlike previous methods that study discrete distributions, we focus on continuous distributions. We propose a relaxation of continuous sampling by formulating the target distribution as a mixture distribution.

## 5   Experiments

We validate the benefits of the proposed sampling-argmax with experiments on a variety of localization tasks, including human pose estimation, retina segmentation and object keypoint estimation.

Additional experiments on facial landmark localization are provided in appendix. Sampling-argmax is compared with the conventional soft-argmax and the variants that using additional auxiliary loss [28]. Training details of all tasks are provided in the supplemental material.

## 5.1   2D Human Pose Estimation from RGB

We first evaluate the proposed sampling-argmax in 2D human pose estimation. In 2D human pose estimation, the probability map is a typical representation to localize body keypoints. The experiments are conducted on the large-scale in-the-wild 2D human pose benchmark – COCO Keypoint [19]. Significant progress has been achieved in this field [37, 33, 27, 25]. We adopt the standard model SimplePose [37] for experiments. We follow the standard metric of COCO Keypoint and use mAP over 10 OKS (object keypoint similarity) thresholds for evaluation.

As shown in Table 1, the proposed sampling-argmax significantly outperforms the soft-argmax operation and its variants. Soft, Soft w/V.R. and Soft w/D.R correspond to conventional soft-argmax, soft-argmax with variance regularization and distribution regularization, respectively. Samp. Uni., Tri. and Gau. correspond to sampling-argmax with uniform, triangular and Gaussian basis, respectively. The triangular basis brings **5.3** mAP improvement (relative **8.2**%) to the original soft-argmax operation. Besides, we find the auxiliary losses degrade the model performance in COCO Keypoint.

Table 1: Quantitative results on COCO Keypoint.

|  | Soft | Soft w/ V.R. | Soft w/ D.R. | Samp. Uni. | Samp. Tri. | Samp. Gau. |
|---|---|---|---|---|---|---|
| mAP $\uparrow$ | 64.5 | 60.6 | 55.6 | 68.2 | **69.8** | 68.3 |
| mAP@0.5 $\uparrow$ | 84.7 | 81.5 | 77.8 | 87.2 | **87.9** | 87.3 |
| mAP@0.75 $\uparrow$ | 70.9 | 65.7 | 60.8 | 75.0 | **76.2** | 75.2 |

**Number of Samples.**   In our method, the differentiable sampling process is utilized to approximate the expectation of the error. As the number of samples increases, the approximation will be closer to the underlying expectation. To study how the number of samples affects the final results, we compare the performance of the models that trained with different numbers of samples. In Table 2, we report the results with $N_s = \{1, 5, 10, 30, 50\}$. It shows that a large number of samples might improve the performance but not necessary. Training the model with only one sample can still obtain high performance while saving computation resources.

Table 2: Comparison of different sample numbers.

| $N_s$ | 1 | 5 | 10 | 30 | 50 |
|---|---|---|---|---|---|
| Samp. Uni. | 67.8 | 67.8 | 67.9 | 68.2 | 68.1 |
| Samp. Tri. | 69.7 | 69.7 | 69.6 | **69.8** | **69.8** |
| Samp. Gau. | 68.1 | 68.1 | 68.2 | 68.3 | 68.3 |

**Correlation with Prediction Correctness.**   For a well-calibrated probability map, the shape of the map could reflect the uncertainty of the regression output. When encountering challenging cases, the probability map would have a large variance, resulting in a lower peak value. In other words, the peak value establishes the correlation with the prediction correctness. To demonstrate the probability map trained with sampling-argmax is better-calibrated, we calculate the *Pearson correlation coefficient* between the peak value and the prediction correctness. The correctness is represented by the OKS between the predicted pose and the ground-truth pose. Table 3 compares the correlation with prediction correctness among

Table 3: Correlation testing.

| Method | Corr. $\uparrow$ |
|---|---|
| Soft | 0.233 |
| Soft w/ V.R. | 0.158 |
| Soft w/ D.R. | 0.082 |
| Samp. Uni. | 0.394 |
| Samp. Tri. | **0.432** |
| Samp. Gau. | 0.423 |

different methods. It shows that sampling-argmax has a much stronger correlation to the correctness than other methods. Compared to the soft-max operation, sampling-argmax with the triangular bases brings **85.4**% relative improvement. It demonstrates that training with sampling-argmax can obtain a more reliable probability map, which is essential to real-world applications and downstream tasks.

## 5.2  3D Human Pose Estimation from RGB

We further evaluate the proposed sampling-argmax on Human3.6M [10], an indoor benchmark for 3D human pose estimation. The 3D probability map is adopted to represent the likelihoods for joints in the discrete 3D space. We adopt the model architecture of prior work [34]. Following previous methods [29, 34, 26, 16], MPJPE and PA-MPJPE [5] are used as the evaluation metrics. Comparisons with baselines are shown in Table 4. The proposed sampling-argmax provides consistent performance improvements. Different from the experiments on COCO Keypoint, the variance regularization provides performance improvements in Human3.6M.

Table 4: Quantitative results on Human3.6M.

|  | Soft | Soft w/ V.R. | Soft w/ D.R. | Samp. Uni. | Samp. Tri. | Samp. Gau. |
|---|---|---|---|---|---|---|
| MPJPE ↓ | 50.4 | 49.7 | 51.9 | 49.6 | **49.5** | 50.9 |
| PA-MPJPE ↓ | 39.5 | 39.2 | 41.4 | 39.1 | 39.1 | **39.0** |

## 5.3  Retina Segmentation from OCT

Using optical coherence tomography (OCT) to obtain 3D retina images is widely used in the clinic. A major goal of analyzing retinal OCT images is retinal layer segmentation. Previous work [7] proposes a regression method to regress the boundary and obtain the sub-pixel surface positions. One-dimensional probability maps are leveraged to model the position distribution of the surface in each column. In the testing phase, the soft-argmax method is used to infer the final surface positions. The entire surface can be reconstructed by connecting the surface positions in all columns.

The experiments are conducted on the *multiple sclerosis and healthy controls* dataset (MSHC) [8]. Mean absolute distance (MAD) and standard deviation (Std. Dev.) are used as evaluation metrics. Quantitative results are reported in Table 5. It shows that sampling-argmax achieve superior performance to other methods, while the auxiliary losses also provide performance improvements.

Table 5: Quantitative results on MSHC dataset.

|  | Soft | Soft w/ V.R. | Soft w/ D.R. | Samp. Uni. | Samp. Tri. | Samp. Gau. |
|---|---|---|---|---|---|---|
| MAD ↓ | 3.08 | 0.743 | 0.746 | **0.735** | 0.744 | 0.740 |
| Std. Dev. ↓ | 0.281 | 0.114 | 0.108 | 0.101 | **0.100** | 0.104 |

## 5.4  Supervised Object Keypoint Estimation from Point Clouds

Detecting aligned 3D object keypoints from point clouds has a wide range of applications on object tracking, shape retrieval and robotics. Probability maps are adopted to localize the semantic keypoints. Different from the RGB input, the probability map indicates the pointwise score of the input point cloud, not the grid position of an image. The distances between the adjacent point-pairs are different. Besides, point clouds are unordered, and each point has a different number of neighbours. Therefore, it is hard to directly apply the uniform bases or linear interpolation, which requires a constant adjacent distance. Fortunately, the Gaussian basis can be adopted. In the experiment, we set the standard deviation $\sigma$ of the Gaussian bases to $0.01$, which is the average adjacent point distance in the input point clouds. PointNet++ [30] is adopted as the backbone network. The experiments are conducted on the large-scale object keypoint dataset – KeypointNet [39]. The percentage of correct keypoints (PCK) [38] is adopted for evaluation. The error distance threshold is set to $0.01$.

Table 6 shows the quantitative results on 16 categories. It shows that the proposed sampling-argmax is also effective on the non-grid input data. Table 6 also compare the results of sampling-argmax with different numbers of samples. It is seen that $N_s = 30$ leads to the best average performance.

## 5.5  Unsupervised Object Keypoint Estimation from Point Clouds

We then evaluate the proposed method on object keypoint estimation in the context of unsupervised learning. The autoencoder framework is adopted to estimate the keypoint in an unsupervised manner.

Table 6: Quantitative results of supervised learning on KeypointNet dataset, reported as PCK (higher is better).

| | Air. | Bat. | Bed | Bot. | Cap | Car | Cha. | Gui. | Hel. | Kni. | Lap. | Mot. | Mug | Ska. | Tab. | Ves. | Avg |
|---|---|---|---|---|---|---|---|---|---|---|---|---|---|---|---|---|---|
| Soft | 64.9 | 43.6 | 44.0 | 53.9 | 8.3 | 40.2 | 37.2 | **45.5** | 4.9 | 43.8 | 46.6 | 40.8 | 23.9 | 27.7 | 53.9 | 32.6 | 38.2 |
| Soft w/ V.R. | 64.1 | 41.6 | 39.2 | 53.2 | 12.5 | 38.3 | 37.7 | 44.5 | 3.7 | 39.8 | **52.8** | 44.0 | 24.9 | 25.6 | 54.4 | 30.7 | 37.9 |
| Soft w/ D.R. | 63.2 | 42.7 | 43.9 | 55.8 | 16.7 | 42.2 | 38.6 | 43.2 | 4.9 | 42.4 | 48.9 | 41.9 | 26.8 | 28.2 | 54.0 | 30.3 | 39.0 |
| Samp. Gau. ($N_s = 1$) | 65.0 | 43.0 | 41.2 | 53.6 | 6.2 | 43.4 | 38.7 | 42.5 | **6.2** | 45.4 | 50.6 | 43.5 | 26.3 | **37.5** | 51.6 | **33.3** | 39.3 |
| Samp. Gau. ($N_s = 5$) | **65.1** | 42.4 | 43.8 | 54.7 | 12.5 | 43.2 | 37.1 | 44.6 | 1.9 | 45.4 | 46.6 | 44.7 | 29.7 | 26.7 | **54.6** | 31.4 | 39.0 |
| Samp. Gau. ($N_s = 10$) | 64.0 | **45.5** | 41.7 | **58.6** | **20.8** | 40.9 | 37.0 | 43.4 | 3.7 | 45.7 | 48.3 | **46.4** | 18.2 | 34.4 | 53.5 | 32.3 | 39.7 |
| Samp. Gau. ($N_s = 30$) | 64.3 | 45.1 | **47.5** | 58.4 | 6.2 | **44.6** | **39.2** | 45.4 | **6.2** | **45.8** | 48.7 | 43.4 | **29.9** | 30.4 | 54.1 | 28.8 | **39.9** |

The encoder first estimates the 3D keypoints, and the decoder reconstructs the object point clouds from the estimated keypoints. We follow the state-of-the-art method [31] that generates 3D keypoints with the soft-argmax operation for differentiable and end-to-end learning. The soft-argmax is replaced with sampling-argmax, where the Gaussian bases with the standard deviation $\sigma = 0.01$ are used.

The experiments are conducted on KeypointNet [39]. Unlike supervised learning, the semantic of each predicted keypoint is unknown in unsupervised methods. Therefore, the PCK metric is not applicable. For evaluation, we adopt the dual alignment score (DAS) following the previous method [31]. Table 7 reports the performance comparison with other methods.

Table 7: Quantitative results of unsupervised learning on KeypointNet dataset, reported as DAS (higher is better).

| | Air. | Bat. | Bed | Bot. | Cap | Car | Cha. | Gui. | Hel. | Kni. | Lap. | Mot. | Mug | Ska. | Tab. | Ves. | Avg |
|---|---|---|---|---|---|---|---|---|---|---|---|---|---|---|---|---|---|
| Soft | 69.1 | 56.2 | 58.0 | 45.4 | 59.1 | **70.2** | 76.8 | 34.1 | 55.7 | 50.0 | 91.5 | 53.4 | 52.2 | 65.7 | 72.5 | 35.8 | 59.1 |
| Soft w/ V.R. | 72.0 | 55.4 | 57.4 | **52.8** | 54.7 | 63.4 | 70.9 | **56.1** | 61.6 | 50.3 | 82.4 | 59.8 | **71.7** | 65.3 | **85.1** | 38.1 | 62.3 |
| Soft w/ D.R. | 47.9 | 35.5 | 47.3 | 46.1 | 58.3 | 65.5 | 60.9 | 35.3 | 47.6 | **69.3** | 64.1 | 55.0 | 45.9 | 44.2 | 57.6 | 28.8 | 50.6 |
| Samp. Gau. ($N_s = 1$) | **73.9** | 53.8 | **63.5** | 43.9 | **67.0** | 69.3 | 77.7 | 46.6 | 59.1 | 55.9 | 87.8 | 59.0 | 67.0 | 66.2 | 80.3 | 36.4 | 62.9 |
| Samp. Gau. ($N_s = 5$) | 73.1 | 54.0 | 61.9 | 48.4 | 64.4 | 67.0 | 81.1 | 50.7 | 55.2 | 50.1 | 87.5 | 58.2 | 58.9 | 65.9 | 77.9 | **41.2** | 62.2 |
| Samp. Gau. ($N_s = 10$) | **73.9** | **58.8** | 61.7 | 46.2 | 60.9 | 68.6 | 72.0 | 53.6 | 56.5 | 48.1 | **91.6** | **59.8** | 68.8 | 65.8 | 83.5 | 34.9 | 62.8 |
| Samp. Gau. ($N_s = 30$) | 71.2 | 56.7 | 60.0 | 51.0 | 58.4 | 64.1 | **83.8** | 47.6 | **61.8** | 47.8 | 91.3 | 55.5 | 68.5 | **70.6** | 81.7 | 37.5 | **63.0** |

## 5.6 Discussion

Although the variants of soft-argmax can bring improvements in some cases, they need laborious tuning of parameters, such as the weight of the regularization term and the variance of the target distribution. The best parameters for different tasks are different. Besides, the best parameters for variance regularization and distribution regularization is also different, which increases the effort needed for the process of parameters tunning. In our experiment, we tune the loss weight ranging from 0.1 to 10 and the variance ranging from 1 to 5 for each task. After laborious tuning, the performances of these variants are still not consistent across different tasks and they are inferior to the performance of our method, while our method is out-of-the-box and free from parameters tuning. Therefore, we think our method is effective and general to different cases.

In addition to a more accurate localization performance, sampling-argmax can predict well-calibrated probability maps and provide more reliable confidence scores. COCO Keypoint uses the mAP metric to evaluate multi-person pose estimation. Thus reliable confidence scores could also improve the performance. In other datasets, the metric only reflects the localization performance and ignore the importance of confidence scores. In many real-world applications and downstream tasks, a reliable confidence score is very important and necessary.

## 6 Conclusion

In this paper, we propose *sampling-argmax*, an operation for improving the detection-based localization. Sampling-argmax implicitly imposes shape constraints to the predicted probability map by optimizing "the expectation of error". With the continuous formulation and differentiable sampling, sampling-argmax can seamlessly replace the conventional soft-argmax operation. We show that sampling-argmax is effective and flexible by conducting comprehensive experiments on various localization tasks.

**Funding** Funding in direct support of this work: the National key R&D Program of China, No. 2017YFA0700800, National Natural Science Foundation of China under Grants 61772332 and Shanghai Qi Zhi Institute, SHEITC (018-RGZN-02046).

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
