# Localization with Sampling-Argmax
## *Supplementary material*

**Jiefeng Li    Tong Chen    Ruiqi Shi    Yujing Lou    Yong-Lu Li    Cewu Lu**
Shanghai Jiao Tong University
{ljf_likit,chentong1023, gzfoxie, louyujing, yonglu_li, lucewu}@sjtu.edu.cn

In the supplemental document, we elaborate on the training settings (Appendix A), the broader impact of our work (Appendix B), limitation and future work (Appendix C), descriptions of the utilized datasets (Appendix D), experiments on facial landmark localization (Appendix E), comparison between the learned distribution of soft-argmax and sampling-argmax(Appendix F), and qualitative results (Appendix G).

## A    Training Details

**2D Human Pose Estimation from RGB**    We adopt SimplePose [17] for experiments. The model is trained and evaluated on COCO Keypoint [13]. ResNet-50 [9] is adopted as the backbone network. The input image is resized to $256 \times 192$. The learning rate is set to $1 \times 10^{-3}$ at first and reduced by a factor of 10 at the 90th epoch and the 120th epoch. We use the Adam solver and train for 140 epochs, with a mini-batch size of 32 per GPU and 8 1080Ti GPUs in total. For comparison with the auxiliary losses, we set the target variance $\sigma_t^2$ to 4, the loss weight of variance regularization to 1, and the loss weight of distributions regularization to 0.1 to achieve the best results after tuning.

**3D Human Pose Estimation from RGB**    We follow the model architecture of Integral Pose [16]. ResNet-50 [9] is adopted as the backbone network. The input image is resized to $256 \times 256$. The learning rate is set to $1 \times 10^{-3}$ at first and reduced by a factor of 10 at the 90th and 120th epoch. We use the Adam solver and train for 140 epochs, with a mini-batch size of 16 per GPU and 8 1080Ti GPUs in total. Following the settings of previous works [16, 14], we mix Human3.6M and MPII [8] data for training. Each mini-batch consists of half 2D and half 3D samples. Five subjects (S1, S5, S6, S7, S8) are used for training and two subjects (S9, S11) for evaluation. We set the target variance $\sigma_t^2$ to 4, the loss weight of variance regularization to 1, and the loss weight of distributions regularization to 0.1 to achieve the best results after tuning.

**Retina Segmentation from OCT**    We follow the model architecture of [10]. The input image is resized to $128 \times 1024$. The learning rate is set to $1 \times 10^{-4}$ at first and reduced by a factor of 10 at the 10th and the 20th epoch. We use the Adam solver and train for 30 epochs, with a mini-batch size of 2 and 1 GPU. The split of training, validation and test sets follows the settings of the previous method [10]. We set the target variance $\sigma_t^2$ to 4, the loss weight of variance regularization to 1, and the loss weight of distributions regularization to 1 to achieve the best results after tuning.

**Supervised Object Keypoint Estimation from Point Clouds**    We adopt PointNet++ [15] as the backbone network. The output of the last layer is a per-point probability map for each keypoint. The input point cloud consists of 2048 points represented by their Euclidean coordinates sampled from a normalized object, and the indexes of keypoints are given. The learning rate is set to $1 \times 10^{-3}$ and halved every 10 epochs. We use Adam solver and train for 100 epochs with a mini-batch size of 8 on one GPU for each category. We set the target variance $\sigma_t^2$ to 4, the loss weight of variance regularization to 1, and the loss weight of distributions regularization to 0.01 to achieve the best results after tuning.

35th Conference on Neural Information Processing Systems (NeurIPS 2021), Sydney, Australia.

**Unsupervised Object Keypoint Estimation from Point Clouds**     The learning rate is set to $1 \times 10^{-3}$ and halved every 10 epochs. We use the Adam solver and train for 50 epochs, with a mini-batch size of 8 and one GPU for each category. We set the target variance $\sigma_t^2$ to 4, the loss weight of variance regularization to 1, and the loss weight of distributions regularization to 0.01 to achieve the best results after tuning.

**Facial Landmark Localization from RGB**     ResNet-18 [9] is adopted as the backbone network. The head network consists of 3 deconvolution layers and a $1 \times 1$ convolution layer. The input image is resized to $256 \times 256$. The learning rate is set to $1 \times 10^{-3}$ at first and reduced by a factor of 10 at the 10th and 20th epoch. We use the Adam solver and train for 30 epochs, which a mini-batch size of 32 and 4 GPUs in total. We set the target variance $\sigma_t^2$ to 4, the loss weight of variance regularization to 1, and the loss weight of distributions regularization to 0.1 to achieve the best results after tuning.

## B    Broader Impact

In this work, we propose sampling-argmax to improve the ability of machines to understand target positions in input data. Current methods usually adopt computationally expensive models to improve the localization accuracy, which could cost many financial and environmental resources. We partly alleviate this issue by presenting a simple yet effective method.

Furthermore, our method is an improvement of existing capabilities but does not introduce a radically new capability in machine learning. Thus our contribution is unlikely to facilitate misuse of technology that is already available to anyone.

## C    Limitation and Future Work

In our method, the underlying density function of the target position is approximated by a mixture of sub-distributions. By comparing the performance of the three proposed bases, we see that a more accurate reconstruction of the underlying function leads to better results. Theoretically, the underlying density function cannot be perfectly reconstructed since the proposed basis distributions are fixed. To address this limitation, learnable sub-distributions could be adopted in future works. For example, *normalizing flow* models can be leveraged to predict sub-distribution at each position according to the corresponding features. In this way, the sub-distributions are no longer fixed, and the mixture distribution has the potential to precisely reconstruct the underlying distribution and further improve the model performance.

## D    Data Acquisition

In our experiments, we use five different datasets, including COCO Keypoint [13], Human3.6M [12], MSHC [11], KeypointNet [18] and MTFL [19]. These public datasets do not contain personally identifiable information or offensive content.

**COCO Keypoint**     COCO Keypoint dataset is licensed under the Creative Commons Attribution 4.0 License [2]. The images and annotations are publicly available. We download the images and annotations from its official website [1].

**Human3.6M**     Human3.6M dataset is licensed under [5]. To obtain the data, we register and download it from its official website [4].

**MSHC**     MSHC dataset is publicly available, and no license is specified. We download the data from its official website [7].

**KeypointNet**     KeypointNet dataset is publicly available, and no license is specified. We download the data from its official website [6].

**MTFL**     MTFL dataset is publicly available, and no license is specified. We download the data from its official website [3].

# E    Facial Landmark Localization from RGB

We further evaluate the proposed sampling-argmax on the facial landmark localization dataset MTFL [19]. Absolute error and relative error (normalized by the two-eye distance) are adopted as evaluation metrics. Quantitative results are reported in Table 1. Consistent with the experiments on other tasks, sampling-argmax provides performance improvement to facial landmark localization.

Table 1: Quantitative results on MTFL dataset.

|  | Soft | Soft w/ V.R. | Soft w/ D.R. | Samp. Uni. | Samp. Tri. | Samp. Gau. |
|---|---|---|---|---|---|---|
| Abs. Err $\downarrow$ | 3.18 | 3.16 | 3.15 | 3.00 | 2.98 | **2.94** |
| Rel. Err $\downarrow$ | 7.25 | 7.22 | 7.20 | 6.86 | **6.82** | 6.96 |

# F    Visualization of learned probability maps

We show the predicted probability maps of soft-argmax and sampling-argmax in Figure 1. It shows that soft-argmax is prone to predict multi-modal distribution, while the proposed sampling-argmax predicts better-calibrated probability maps.

# G    Qualitative Results

Qualitative results on six tasks are shown in Figure 2, 3, 4, 5, 6 and 7.

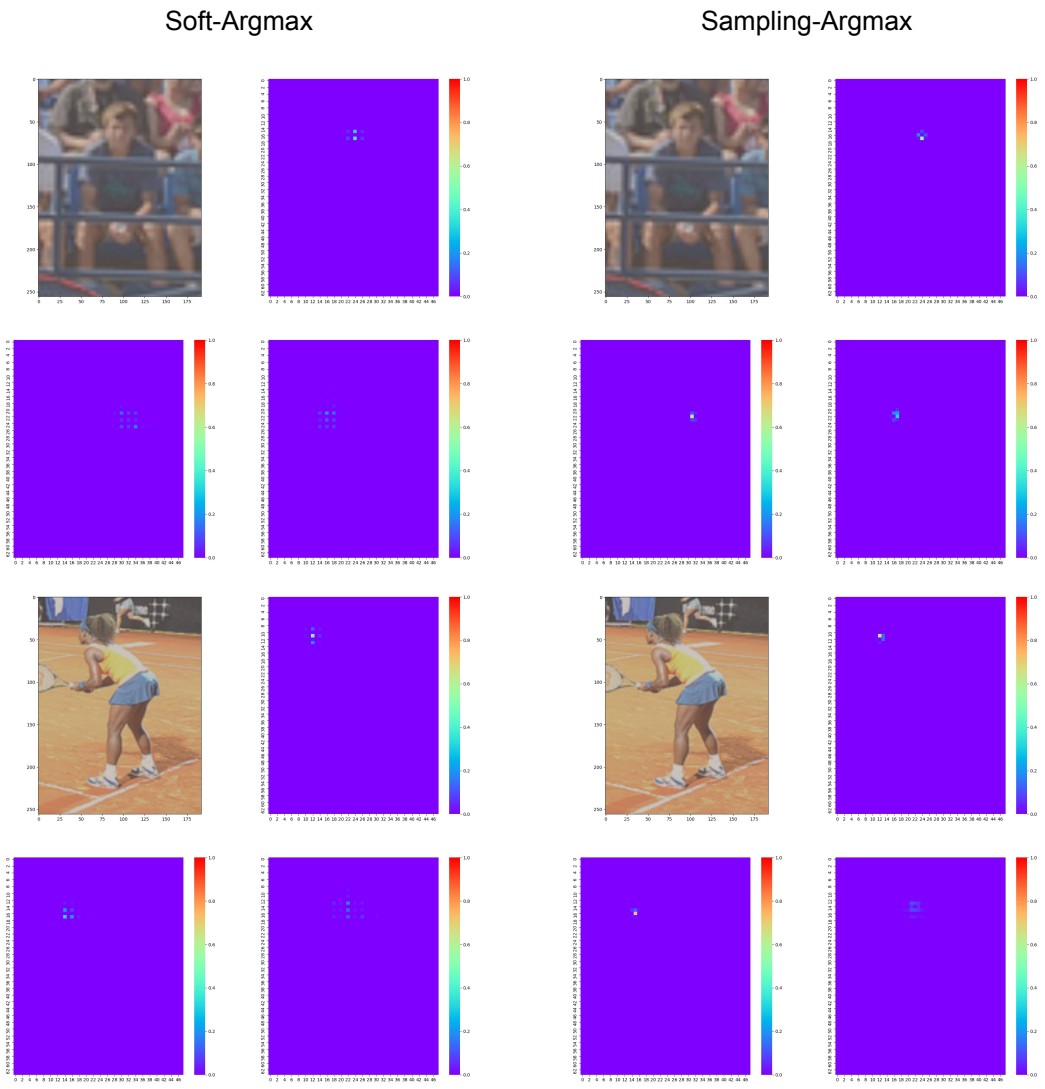

Figure 1: **Visualization** of the learned distribution. **Left**: Soft-Argmax. **Right**: Sampling-Argmax.

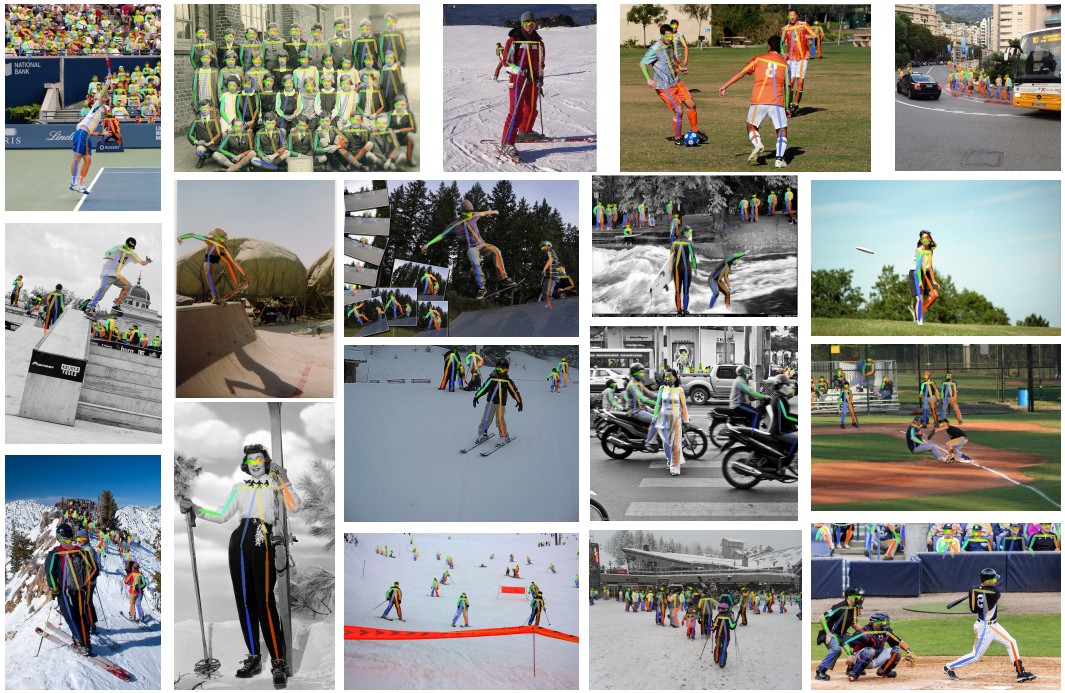

Figure 2: **Qualitative** results of 2D human pose estimation on COCO Keypoint.

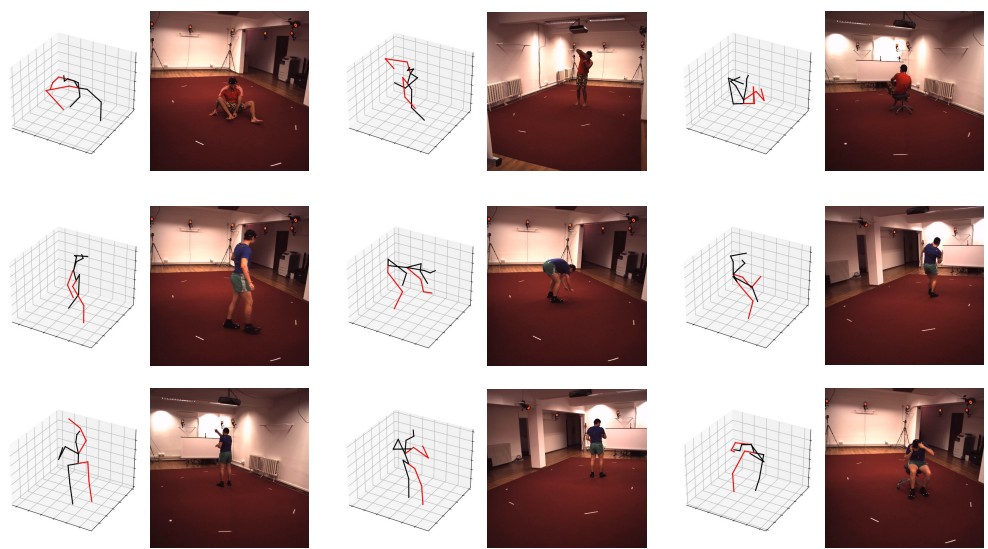

Figure 3: **Qualitative** results of 3D human pose estimation on Human3.6M.

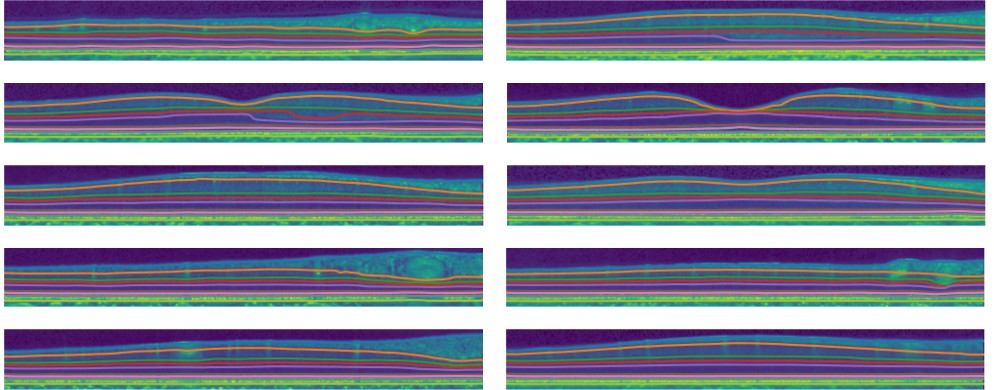

Figure 4: **Qualitative** results of retina segmentation on MSHC.

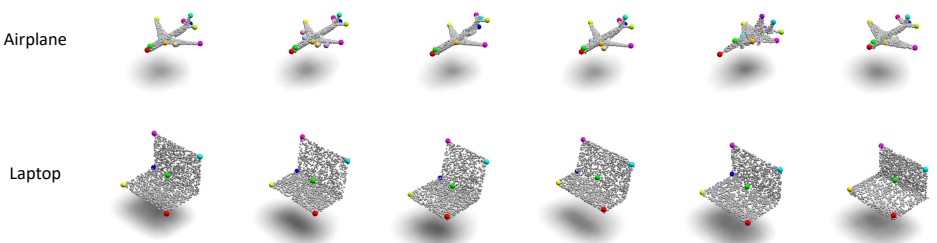

Figure 5: **Qualitative** results of supervised model on KeypointNet.

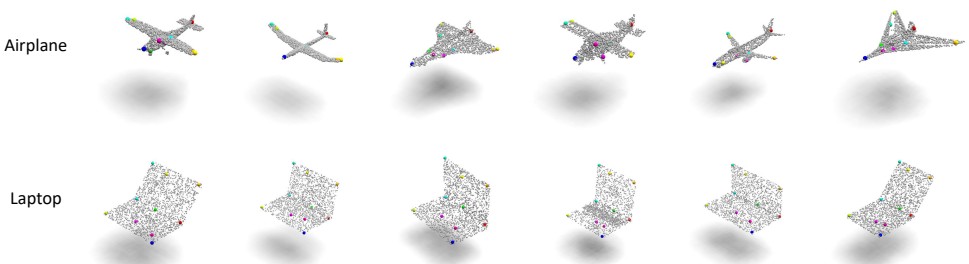

Figure 6: **Qualitative** results of unsupervised model on KeypointNet.

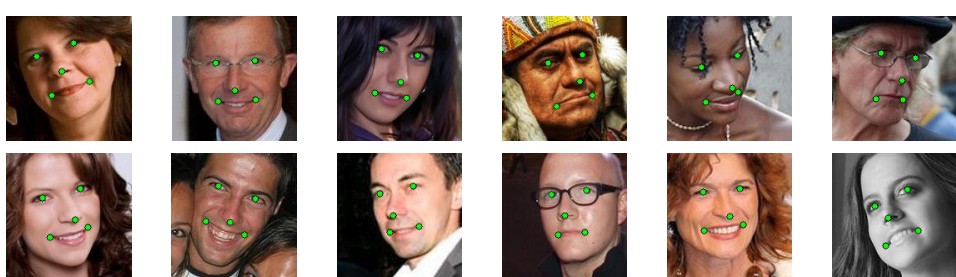

Figure 7: **Qualitative** results of facial landmark localization on MTFL.