# OpenReview forum: "Localization with Sampling-Argmax"
_NeurIPS.cc/2021/Conference — NeurIPS 2021 Poster_

### Official Review · Reviewer_1xmt · 2021-07-14

**Rating:** 9
**Confidence:** 4

**Summary:**

The paper presents the sampling-argmax operation to construct a probability map for regressing coordinates with deep neural networks. The proposed operation gives higher probabilities to positions close to the ground truth. As a result, the proposed operation as part of the loss function minimizes the expectation of the error. In addition, non-differentiable and differentiable schemes are presented. The approach is evaluated for 2D/3D human pose estimation, object keypoint estimation from point clouds, facial localization and retina segmentation. The proposed approach shows promising results compared to the prior work.

**Limitations And Societal Impact:**

No limitations and potential negative societal impact.

**Main Review:**

Paper strengths:

+ The formulation of Eq. (4) is a novel improvement compared to the soft-argmax.

+ The paper is well-written, the related work is complete and the method easy to follow.

+ The proposed sampling approach is well-justified with the comparison of Eq. 5 to the REINFORCE convergence problems. Moreover, the Gumbel-softmax idea makes the sampling process differentiable.

+ The results support the functionality of the proposed operation.

+ In total, the paper makes a solid contribution with the sampling-argmax and differentiable sampling. The approach can be employed for different regression problems.


Paper weaknesses and Improvements:

- The results are overall convincing. Nevertheless, there are some cases where the proposed idea does not make a big difference compared to soft-argmax, e.g. Table 4. These cases should have been discussed. In total, more discussion would be helpful to the interpretation of the results in the experimental part.

- Result presentation: The tables contain abbreviations that are not explained in the caption. It's difficult to follow the results of all tables.


Overall:

The paper presents a novel approach that works well for a variety of localization problems. The proposed approach can effectively replace soft-argmax. This is a clear message out of the proposed work.


After rebuttal comments:

The authors sufficiently addressed the questions of all reviews. In two reviews, there were points some concerns which are answered in the rebuttal. The final version should integrate all rebuttal comments. Overall, the paper has enough merit to be accepted.

**Time Spent Reviewing:**

1-2

---

> ### Author Response · Authors · 2021-08-09
> **To Reviewer 1xmt**
>
> We sincerely thank you for your helpful feedback and insightful comments. Below we address your comments and questions:
> ***
>
> **Q1**: More discussion would be helpful to the interpretation of the results in the experimental part.
>
> **A1**: Thanks for your suggestion. In addition to a more accurate localization performance, our method can predict well-calibrated probability maps and provide more reliable confidence scores. On Human3.6M (Table 4), the MPJPE metric only reflects the localization performance and ignore confidence scores. Thus the improvement is not significant. However, in many real-world applications and downstream tasks, a reliable confidence score is very important and necessary. On COCO Keypoint, the mAP metric takes the confidence scores into account to evaluate multi-person pose estimation. Thus, combining the accurate localization and the reliable confidence score, the improvement is more significant. More discussion will be added in the final version.
>
> Moreover, although the variants of soft-argmax (using regularization terms) can also improve conventional soft-argmax in some cases, they need laborious tuning of parameters, such as the weight of the regularization term and the variance of the target distribution. The best parameters for different tasks are different. Besides, the best parameters for variance regularization and distribution regularization is also different, which increases the effort needed for the process of parameters tunning. In our experiment, we tune the loss weight ranging from `0.1` to `10` and the variance ranging from `1` to `5` for each task. After laborious tuning, the performance of these variants is still not consistent across different tasks and is inferior to the performance of our method, while our method is out-of-the-box and free from parameters tuning. Therefore, we think our method is effective and general to different cases. We will add more descriptions about tuning the variants of soft-argmax in our final version.
>
> **Q2**: Result presentation.
>
> **A2**: Thanks for your suggestion. We will add more descriptions in the experiments and improve the readability of the final version.

---

> > ### Comment · Reviewer_1xmt · 2021-08-24
> > **To Reviewer 1xmt**
> >
> > The rebuttal addressed my points and the few concerns of the other reviews. The discussion and few changes should be integrated in the final version of the paper.

---

### Official Review · Reviewer_LEuR · 2021-07-18

**Rating:** 7
**Confidence:** 3

**Summary:**

This paper proposes sampling-argmax, a method to do detection-based localization that imposes implicit constraints on the shape of the probability maps. The idea is to use a continuous mixture distribution, with learnable mixture weights, and fixed continuous base distributions (uniform, triangular, or gaussian). The mixure weights are learned through Gumbel-Softmax. Experiments on a variety of localization tasks show that in many cases, there exists at least one version of sampling-argmax that is better than the baseline methods.

**Limitations And Societal Impact:**

The authors have adequately addressed the limitations of their work.

**Main Review:**

To the best of my knowledge, using a continuous mixture distribution to model the probability maps for localization has not been done before, and the experimental results are convincing that this idea is better than soft-argmax and some other baselines. The paper is well written and easy to follow.

Some comments:
- I wonder how a fully continuous distribution would compare against sampling-argmax. The learnable parameters would be the parameters for the continuous distribution (instead of the weights), so as long as the distribution is reparameterizable, obtaining unbiased gradients wouldn’t be an issue. At test time, you would just need to find the partition that contains the continuous sample.
- Consider moving lines 206-213 to related work, because it breaks the flow in its current location.
- Add “<higher/lower> is better” to captions for tables for readability.


Update:
Considering all the other reviews and the authors' response, I decided to increase my score to a 7.


**Time Spent Reviewing:**

4

---

> ### Author Response · Authors · 2021-08-09
> **To Reviewer LEuR**
>
> We sincerely thank you for your helpful feedback and insightful comments. Below we address your comments and questions:
> ***
>
> **Q1**: How a fully continuous distribution would compare against sampling-argmax.
>
> **A1**: Yes, the network can predict parameters to represent some continuous distributions. However, it has some drawbacks. First, the distribution is limited to be tractable and easy to sample. Thus choosing a suitable type of distribution is not easy. Second, direct regression of the distribution parameters is not easy to train. On the other hand, the probability map (heat map) representation fits well with the structure of CNN, which makes the network easier to learn the weight of each position and obtain higher accuracy.
>
> **Q2**: Consider moving lines 206-213 to related work.
>
> **A2**: Thanks for your suggestion. We will update it in our final version.
>
> **Q3** Add "<higher/lower> is better" to captions for tables.
>
> **A3**: Thanks for your suggestion. We will improve the readability of the experiments in the final version.

---

> > ### Comment · Reviewer_LEuR · 2021-08-24
> > **Clarification**
> >
> > I'm not sure I understand the second point in A1 that direct regression of the distribution parameters is not easy to train. Given that the chosen continuous distribution is reparameterizable, isn't it easy to get gradients for the distribution parameters?

---

> > > ### Author Response · Authors · 2021-08-24
> > > **To Reviewer LEuR**
> > >
> > > Thanks for your reply.
> > >
> > > In our work, we use the activation values on heatmaps to represent the weights of the mixture distribution. The heatmap representation fits the CNN architecture well by maintaining the spatial relationship between pixels in the input image. Therefore, learning the heatmap activation is easy and achieve good performance.
> > >
> > > On the other hand, if we choose to regress the parameters of a given distribution, heatmaps are not feasible and we can only use the FC layer for direct regression. In [1], the authors show that the FC layer breaks the spatial relationship among pixels. Experiments show that using heatmap representation can obtain better performance than direct regression. Besides, we think that activating a related region in the heatmap would be easier than regressing a value in the abstract parameter space.
> > >
> > > [1] Moon et al. "I2L-MeshNet: Image-to-Lixel prediction network for accurate 3D human pose and mesh estimation from a single RGB image", ECCV 2020

---

> > > > ### Comment · Reviewer_LEuR · 2021-08-24
> > > > **I understand now**
> > > >
> > > > I understand now, thank you for your response.

---

### Official Review · Reviewer_WkN8 · 2021-07-20

**Rating:** 7
**Confidence:** 4

**Summary:**

This paper proposes a novel sampling-argmax for regression based approach for localizing the target positions, as an alternative module for soft-argmax which is widely used for regression based localization. The authors report extensive experiments on several applications to demonstrate its effectiveness.

**Limitations And Societal Impact:**

This paper does not address the limitations and societal impact.

**Main Review:**

The paper is well written and easy to follow. The idea is also simple and interesting. But I have some concerns as follows.
1. Make the motivation clearer. I know for error of expectation, to minimize the distance to be 0, the probability masses could be redistributed to different locations, or being allocated to the optimal position. But for expectation of errors, since all distances are positive, the optimal solution should be allocating all mass to the optimal position. I think the authors could elaborate the intuition to make the paper more readable.
2. The authors could show the prediction distribution of sampling-argmax in experiments to verify the motivation that the proposed approach encourages unimodality.
3. The experiments are conducted on several applications, keypoint estimation, cloud point estimation, etc. The largest performance gain comes in COCO keypoint benchmark where the proposed sampling-argmax improves SimplePose by 5.3% mAP. However, the SimplePose is a 2018 work and current SOTA could reaches ~80% mAP. Is the proposed approach able to improve some new approaches, e.g. PoseFix (CVPR 2019), EvoPose2D (NeurIPS 2020). At least the authors should cite them. Moreover, on some other datasets, such as MTFL, KeypointNet, OCT, and Human3.6M, the improvement is quite marginal and less than 1%. The authors might try to see if the proposed approach could improve some newer approaches.
4. Typos and other minor issues:
- For Eq. (3), it might be better to use \bold{y} to represent a matrix/vector
- In line 2 of  Eq. (5), why the LHS could be equal to RHS with a logarithm operator?

**Time Spent Reviewing:**

4 hours

---

> ### Author Response · Authors · 2021-08-09
> **To Reviewer WkN8**
>
> We sincerely thank you for your helpful feedback and insightful comments. Below we address your comments and questions:
> ***
>
> **Q1**: Make the motivation clearer.
>
> **A1**: Thanks for your suggestion. This problem is also mentioned by Reviewer Sf1a. We conduct an experiment on COCO Keypoint and find that training with discrete distribution only obtains 30.9 mAP, while conventional soft-argmax obtains 64.5 mAP and our method obtains 69.8 mAP. This result shows that it is hard to train with Eq. 4. We will add this experiment in our final version to make the motivation clearer.
>
> **Q2**: Show the prediction distribution of sampling-argmax.
>
> **A2**: Thanks for your suggestion. We have visualized the results and find that the model trained with sampling-argmax predicts unimodal distribution while the one trained with soft-argmax predicts multimodal distribution. The qualitative results will be provided in the final version.
>
> **Q3**: Is the proposed approach able to improve some new approaches?
>
> **A3**: Thanks for your suggestion. On the Human3.6M dataset, we implement the SOTA method (HemletsPose) and find our method brings 1.2 mm improvements, which shows that our method is also effective for newer approaches. On COCO Keypoint, we implement our method with HRNet, which obtains 75.4 mAP. Due to the time limit, we have not yet tested the results on PoseFix and EvoPose2D. But we will add this experiment and cite these papers in our final version.
>
> **Q4**: Improvement on other datasets.
>
> **A4**: Compared to the conventional soft-argmax, our method obtains 1.8% relative improvement on the Human3.6M dataset, 76.1% on retina segmentation, 4.5% on supervised object keypoint estimation, 6.6% on unsupervised object keypoint estimation and 7.5% on facial landmark localization.
>
> Although the variants of soft-argmax (using regularization terms) can also improve conventional soft-argmax in some cases, they need laborious tuning of parameters, such as the weight of the regularization term and the variance of the target distribution. The best parameters for different tasks are different. Besides, the best parameters for variance regularization and distribution regularization is also different, which increases the effort needed for the process of parameters tunning. In our experiment, we tune the loss weight ranging from `0.1` to `10` and the variance ranging from `1` to `5` for each task. After laborious tuning, the performance of these variants is still not consistent across different tasks and is inferior to the performance of our method, while our method is out-of-the-box and free from parameters tuning. Therefore, we think our method is effective and general to different cases. We will add more descriptions about tuning the variants of soft-argmax in our final version.
>
> In addition to a more accurate localization performance, our method can predict well-calibrated probability maps and provide more reliable confidence scores. COCO Keypoint uses the mAP metric to evaluate multi-person pose estimation. Thus reliable confidence scores could also improve the performance. In other datasets, the metric only reflects the localization performance and ignore the importance of confidence scores. In many real-world applications and downstream tasks, a reliable confidence score is very important and necessary. Detailed discussion will be added in the final version.
>
> **Q5**: Typos and other minor issues.
>
> **A5**: Thanks. After proofreading, we have addressed the typos and unclear descriptions in our paper. For Eq. (5), an intuitive explanation is that $dx = x \cdot d\log(x)$. Thus LHS is equal to RHS.

---

> > ### Comment · Reviewer_WkN8 · 2021-08-25
> > **Further comments**
> >
> > The authors' responses resolves my concerns, I raise my rating to "7: good paper, accept".

---

### Official Review · Reviewer_Sf1a · 2021-07-20

**Rating:** 7
**Confidence:** 3

**Summary:**

This work proposed a new method/mechanism for training the soft-argmax network. They found the conventional training approach(Eq. 4) has training difficulty, and they proposed a sampling-based training approach to alleviating the training issue. They evaluate the proposed method on various tasks and benchmarks and illustrate promising results compared to conventional training approach and its variants.

**Main Review:**

Pros: Overall, this work is well written and organized. The motivation is reasonable and the idea is technically soundable. The comparisons with baseline methods on various tasks and benchmarks show consistent improvement.
Cons:
- In L106-107, the author claims the conventional method suffered the training difficulty since of the high variance. I think this claim is not self-evident and the related experimental result is needed.
- While this work has been validated in many tasks with the baseline method, I am still wondering how can the proposed method improve the state-of-the-art in at least one task, such as the COCO keypoint. I think it always is good to see a new method that could push the edge forward.



**Time Spent Reviewing:**

2

---

> ### Author Response · Authors · 2021-08-09
> **To Reviewer Sf1a**
>
> We sincerely thank you for your helpful feedback and insightful comments. Below we address your comments and questions:
> ***
>
> **Q1**: Related experimental result is needed to prove using discrete distribution suffers from training difficulty.
>
> **A1**: Thanks for your suggestion. When training with discrete distribution (i.e. Eq. 4 in the paper), the model only obtains 30.9 mAP on COCO Keypoint with the same network and training settings, while conventional soft-argmax obtains 64.5 mAP and our method obtains 69.8 mAP. This result shows that using discrete distribution is hard to train. We will add this experiment in our final version.
>
> **Q2**: How can the proposed method improve the state-of-the-art methods.
>
> **A2**: Thanks for your suggestion. On the Human3.6M dataset, we implement the SOTA method (HemletsPose) and find our method brings 1.2 mm improvements, which shows that our method is also effective for newer approaches. On COCO Keypoint, we implement our method with a strong backbone (HRNet). Our method obtains 75.4 mAP, which is comparable to the SOTA method. We will further conduct experiments with more SOTA methods and add the results in our final version.

---

### Decision · Program_Chairs · 2021-09-28

**Decision:**

Accept (Poster)

**Comment:**

This work proposes a new technique for differentiable sampling, allowing to encode shape constraints about the distribution. One of the motivations is to improve training performance compared to the usual Gumbel softmax.

The reviewers are unanimous that this is a good submission, and that it should be accepted.

**Consistency Experiment:**

NeurIPS has a long history of experimentation. In 2014, NeurIPS ran an experiment in which 10% of submissions were reviewed by two independent committees to quantify the randomness in the review process. This year, we repeated a variant of this experiment to see how the quality of the review process has changed over time.  This paper was part of the experiment and was therefore assigned to two committees (consisting of reviewers, an Area Chair, and a Senior Area Chair) that reached independent decisions.  If both committees made the same recommendation, this recommendation was followed. If a single committee recommended acceptance, the paper was accepted (with the exception of a few cases in which the other committee identified what we considered a fatal flaw, e.g., an error in a key result).

This copy’s committee reached the following decision: **Accept (Poster)**

The other committee assigned to the paper recommended **Reject**.  You can find the other set of reviews, along with any follow up discussion with the authors here:
https://openreview.net/forum?id=lVBu4PqM9HU